# Assessing a bundle of peer counseling, mobile phone messages, and mama kits in promoting timely initiation of and exclusive breastfeeding in Uganda: A cluster randomized controlled study

David Mukunya [1,2,3]*, James K. Tumwine[4,5], Grace Ndeezi[4], Milton W. Musaba[6], Justin Bruno Tongun[7], Josephine Tumuhamye[8], Agnes Napyo[9], Faith Oguttu[1], Daphine Amanya[1], Beatrice Odongkara[10], Vincentina Achora[11], Thorkild Tylleskar[3], Victoria Nankabirwa[3,12]

1 Department of Community and Public Health, Busitema University, Mbale, Uganda, 2 Department of Research, Nikao Medical Center, Kampala, Uganda, 3 Centre for Intervention Science and Maternal Child Health (CISMAC), Centre for International Health, University of Bergen, Bergen, Norway, 4 Department of Paediatrics and Child Health, School of Medicine, Makerere University College of Health Sciences, Kampala, Uganda, 5 Department of Paediatrics and Child Health, School of Medicine, Kabale University, Kampala, Uganda, 6 Busitema University Faculty of Health Sciences, Department of Obstetrics and Gynaecology, Mbale, Uganda, 7 Department of Paediatrics and Child Health, University of Juba, Juba, South Sudan, 8 Department of Research, Makerere University Hospital, Kampala, Uganda, 9 Department of Nursing, School of Medicine, Kabale University, Kabale, Uganda, 10 Department of Paediatrics and Child Health, Gulu University, Gulu, Uganda, 11 Department of Obstetrics and Gynaecology, Gulu University, Gulu, Uganda, 12 Department of Epidemiology and Biostatistics, School of Public Health, Makerere University College of Health Sciences, Kampala, Uganda

* zebdaevid@gmail.com

## Abstract

### Background

Timely initiation of and exclusive breastfeeding have been recommended as key interventions to enable countries to attain the sustainable development target of reducing neonatal mortality to no more than 12 deaths per 1000 live births and to reduce mortality of children under 5 years to no more than 25 deaths per 1000 live births.

### Methods

We conducted a cluster randomized controlled trial with the main objective to assess the effect of an integrated package consisting of: peer counseling, mobile phone messages, and mama kits on promoting health facility births between January 2018 and February 2019, in Lira district, Northern Uganda. In this article, we assessed the effect of the intervention on our two secondary objectives: timely initiation of and exclusivity of breastfeeding. We used a generalized estimation equation of the Poisson family, with a log or identity link, taking clustering into account to estimate prevalence ratios and prevalence differences.

**Data Availability Statement:** All relevant data are within the manuscript and its Supporting Information files.

**Funding:** We acknowledge the Survival Pluss project (no. UGA-13-0030) at Makerere University. Survival Pluss project was funded under NORHED by Norad; Norway. The funders had no role in study design, data collection and analysis, decision to publish, or preparation of the manuscript.

**Competing interests:** The authors have declared that no competing interests exist.

## Results

A total of 64% (594/926) of participants in the intervention arm initiated breastfeeding within the first hour after birth compared to 60% (493/829) in the control arm. The proportion of participants in the intervention arm that initiated breastfeeding within the first hour of life did not significantly differ from that in the control arm [Prevalence Ratio (PR) 1.08 (0.97 to 1.21)] and [Prevalence Difference (PD) 0.05 (-0.02 to 0.12)]. When we restricted the analysis to only mothers who decided on when to breastfeed, there was some evidence of intervention effectiveness [PR 1.20, 95% CI (0.99–1.5)]. In the intervention arm, 89% (804/904) of participants exclusively breastfed their infants in the first month of life compared to 81% (656/813) in the control arm. Participants in the intervention arm were 10% more likely to have exclusively breastfed in the preceding 24 hours compared to mothers in the control arm [PR 1.10 (1.04 to 1.17)] and [PD 0.08 (0.04 to 0.13)], and 16% more likely to have exclusively breastfed since birth compared to mothers in the control arm [PR 1.16 (1.03 to 1.30)] and [PD 0.12 (0.03 to 0.20)].

## Conclusion

The intervention improved the proportion of mothers who practiced exclusive breastfeeding in the first month of life, but did not increase the proportion of mothers who initiated breastfeeding in the first hour of life. Future breastfeeding promotion interventions should consider including a health facility component and improving maternal autonomy to promote timely initiation of breastfeeding.

## Introduction

Uganda aims to reduce under-5 mortality to less than 25 deaths per 1,000 live births, and neonatal mortality to no more than 12 deaths per 1,000 live births by 2030 [1,2]. Currently, under-5 mortality in Uganda is estimated at 52 deaths per 1,000 live births and neonatal mortality is estimated at 22 deaths per 1,000 live births [3]. In order to achieve the targets above, interventions that promote timely initiation of and exclusive breastfeeding have been recommended [4–6].

Timely initiation of breastfeeding and exclusive breastfeeding independently reduce both severe morbidity and mortality among children under the age of 5 years [5,7–11]. Children who are breastfed earlier are less likely to suffer severe neonatal illness and experience fewer episodes of diarrhea and other infectious diseases [12,13]. This can be attributed to the ingestion of colostrum, which has immunological, anti-inflammatory, and nutritional benefits [4,7]. In addition, children who are breastfed early benefit from their mothers' warmth and bonding, and they are more likely to be exclusively breastfed and to be breastfed for longer periods [4,14,15]. Furthermore, breastfeeding confers long-term benefits to the immune system of newborns [16]. There is evidence that babies ingest their mothers' immune cells during breastfeeding and these cells gain access to the newborn's lymphoid tissues during the immediate postpartum period when the newborn's intestinal walls are highly porous [17–19]. The period of high porosity is thought to last for a short while after birth, creating a "crucial gap period". Timely initiation of breastfeeding enables the utilization of this crucial period and results in lifelong benefits to the newborn [17]. Despite these advantages, the prevalence of timely

initiation of breastfeeding and exclusive breastfeeding in sub-Saharan Africa, and the rest of the world remains too low [20].

Only 63% of women in sub-Saharan Africa practice timely initiation of breastfeeding [21]. In 2016, 66% of mothers in Uganda reported to have initiated breastfeeding within the first one hour of giving birth [22]. However, the prevalence of timely initiation of breastfeeding among women in Northern Uganda has been reported to be much lower at 51.8% [23]. This finding was similar to that of a hospital-based study in South Sudan, which showed that only half of the mothers practiced timely initiation of breastfeeding [24]. The scale-up of timely initiation of breastfeeding is listed as a priority in the reduction of child mortality [25]. To scale up the timely initiation of breastfeeding, the World Health Organization's guidelines on the promotion of breastfeeding calls for the involvement of peer counselors and family members in promoting breastfeeding [26].

A Cochrane review assessing interventions for promoting timely initiation of breastfeeding showed that available evidence was of low quality, and lacked generalizability to low-income countries [27]. Studies have shown that peer counselors and mobile phone messages improve adherence to recommended breastfeeding practices [28–32]. However, no study has assessed the effect of more than a single intervention to promote timely initiation of breastfeeding.

We conducted a cluster randomized controlled trial (ClinicalTrials.gov NCT02605369) with the primary objective to evaluate the impact of an integrated intervention (counseling by peer counselors, mobile phone messaging and distribution of mama kits) on the proportion of mothers who gave birth in health facilities [33]. This article presents the analysis of the effect of the intervention on the secondary objective: to assess for timely initiation of and exclusivity of breastfeeding.

## Methods

### Study design and participants

The trial was conducted in Lira district, Northern Uganda between January 2018 and February 2019. A map of the study villages is attached as S5. The unit of randomization of the primary trial was a cluster made up of 5 to 10 villages with a population of at least 1000 people. Details about masking and randomization have been reported [33]. We included pregnant women at $\geq$ 28 weeks of gestation or visibly pregnant within the study villages and recruited in the primary study.

### Interventions and standard antenatal care

The intervention was an integrated bundle consisting of peer support by pregnancy buddies, provision of mama kits, and mobile phone messaging administered at the individual and family levels.

**Peer counseling.**   Each village elected their peer counselor during the sensitization meeting. These were literate women of reproductive age (18–45 years), who were trained for 3 days plus a monthly one-day refresher training and feedback session for the entire period of the trial. Training materials used to promote breastfeeding practices were adapted from a similar study in Uganda conducted by the same investigators [28]. Counseling skills were taught by demonstrations and role-plays. In total, 114 peer counsellors were trained. Each peer buddy counselled 5–15 mothers, and each visit would last between 20–60 minutes. After obtaining consent from the participant, at least four peer buddy visits were scheduled. The first visit would take place immediately after recruitment while the next two visits were scheduled to take place at the mother's convenience before delivery. The last visit would take place within the first three days post-partum. Topics discussed in the first three visits included:- (1)

encouraging health facility births, (2) developing a birth preparedness plan, (3) counseling the mother on danger signs in pregnancy, (4) benefits of initiating breastfeeding within the first hour, (5) benefits of exclusive breastfeeding, (6) advantages of colostrum, (7) skin-to-skin care, and (8) the dangers of pre-lacteal feeding. The fourth visit mainly encouraged the mother to go to the health facility for postnatal care within the first week after birth. Counseling took place at the mothers' homes and involved the mother, husband, mother-in-law, and any significant others. Some of the peer counselling messages used include the following;- *"Breastfeeding is the best way. It is safe, prevents illness, and helps your baby grow strong. Let your baby suckle whenever he wants"*, *"Your baby needs nothing else apart from breast milk for the first 6 months. Your milk will contain all the water and goodness he needs"*, *"The first milk is medicine for the baby, don't throw it away. It's not dirty; that is its color"*

**Mobile phone messaging.** Peer buddy counseling was supplemented by mobile phone messaging to the study participant or any other family member, in case the participant did not have a mobile phone. The mobile phone messages contained the same messages discussed in the counseling sessions:- (1) encouraging health facility births, (2) birth preparedness, (3) early initiation of breastfeeding, and skin-to-skin care. The text messaging system was automated with messages being sent weekly until birth. A message was also sent after birth to encourage postnatal health facility visits. The messages were validated and translated into Lango, the local language. The role of mobile phone messages in this study was to reinforce the information offered by peer counsellors and to remind mothers. Some of the mobile phone messages shared were;- *"Welcome to Survival Plus SMS service! We will be sending a weekly message about your pregnancy",*" *Make a plan with your family to put your new baby to the breast in the first hour. Your creamy first milk will protect him from illness"*, *"Your first milk is best for your baby. Let your partner know you want to breastfeed your baby within the first hour. He can support you"*, *"Breastfeeding helps to reduce your bleeding after birth. Go to the clinic if your bleeding becomes heavy, clotted, smelly or you feel faint"*

**Mama kits.** All participants in the intervention clusters were given mama kits. Mama kits are clean delivery kits that contain; gauze, cotton wool, a razor blade, umbilical cord ties, soap, two pairs of sterile gloves, a polythene sheet and a child growth monitoring and immunization card.

Further details about each intervention can be found elsewhere [33]. Participants in the control area received the standard of care according to Ministry of Health guidelines. This consisted of occasional radio health promotional messages, as well as information obtained during the antenatal, natal, and postnatal health facility visits.

## Ethical issues

Ethical approval to conduct the study was obtained from the following bodies: 1) Research and Ethics Committee School of Medicine, Makerere University (SOMREC: REF 2015–121); 2) Uganda National Council of Science and Technology (UNCST: SS 3954); 3) Regional Committees for Medical and Health Research Ethics (REK VEST 2017/2079) and the trial was registered at ClinicalTrial.gov as NCT02605369 and published as well [33]. We also obtained permission from the Ministry of Health and Lira Local Government. Written informed consent was obtained from the respondents in the study. Research assistants were trained on the importance of confidentiality of the information and the right of the respondents to withdraw their participation at any time during the study. At the community level, we obtained permission to include clusters during community sensitization meetings; after which the community members elected recruiters, and peer counselors when applicable from amongst themselves.

## Sample size

The sample size of the parent study was calculated based on the primary objective [34]. We used Stata IC version 14 (StataCorp, College Station, Tx, USA) to verify that the given sample size was sufficient to answer our primary objective with 90% power. A baseline survey conducted between August 2016 and November 2016, indicated that only 51.8% of mothers with infants below 2 years initiated breastfeeding within the first hour after birth [23]. We hypothesized that an intervention consisting of counseling by peer counselors, distribution of mama kits and mobile phone messaging, to pregnant women and their family members and (or) significant others, would increase the proportion of mothers who practiced timely initiation of breastfeeding from 50% to 70%, compared to mothers in the control arm who received the standard antenatal care.

We also assumed an average cluster size of 50 pregnant women; and an intra-cluster correlation coefficient (ICC) of 0.09. To detect a 20% increase in the proportion of mothers initiating breastfeeding within the first hour in the intervention clusters; we needed 15 clusters per arm and a minimum sample size of 750 participants per arm or a total of 1500. This sample size was less than 1800 that had been calculated for the primary objective of the parent study.

## Measurements

**Outcomes.**   The outcomes of the current analysis were timely initiation of breastfeeding and exclusive breastfeeding during the first 28 days of life. Timely initiation of breastfeeding was defined as initiating breastfeeding within the first hour after birth. Research assistants approached participants within 24 hours after birth, and asked them whether they had initiated breastfeeding, and if yes, after how long in minutes and hours. If the mother had not initiated breastfeeding at the time of the first visit, the research assistant returned on day 7 and day 28. Mothers were asked the following questions to assess for the key decision maker in initiating breast feeding. "*Who decided when to initiate breastfeeding*", "*Who decided what to do with the initial breast milk*"

A research assistant visited the mother on day 28 to assess for exclusive breast feeding. Exclusive breast feeding was assessed by 24-hour recall and a one-month recall. A 24-hour recall was done by the research assistant, the mother was asked to list the different food stuffs she had fed the baby in the last 24 hours. A one-month recall was also done and mothers were asked to list the different food stuffs they had fed the baby in the last 28 days.

Exclusive breastfeeding was defined as not giving the baby anything else apart from breast milk, modern medicines (including vaccines and vitamins), and oral rehydration salts for the first 28 days of life.

**Baseline characteristics and potential confounders.**   Data were collected on potential confounders for all participants enrolled in the trial during pregnancy and immediately after birth. These included: maternal age, parity (number of pregnancies carried to $> = 37$ weeks of gestation), maternal education, occupation, antenatal care attendance, maternal morbidity, wealth quintiles, household size, singleton or multiple birth, sex of the infant, birth weight, gestational age, place of birth, and marital status. Wealth quintiles were calculated from an asset-based index using principal component analysis. The following assets and house characteristics were considered:- Cupboard, bicycle, radio, mobile phone, motorcycle, cement floor, iron sheets, burnt bricks, and land ownership.

## Data collection, management and quality control

Data were collected electronically on standardized questionnaires and forms using Open Data Kit (www.opendatakit.org) software installed on android phones. These questionnaires had

inbuilt checks for data accuracy. Participants were interviewed at recruitment to assess eligibility and obtain informed consent. Baseline data were also collected to obtain information on potential confounding factors and socio-demographic factors. Information on timely initiation of breastfeeding was collected on day 1, and that on exclusive breastfeeding was collected on day 28 by a 24-hour recall. In case the mother was absent on day 1 or had not initiated breastfeeding by the time of the interview, subsequent visits on day 7 and day 28 would be used to collect the information. All this was done as stipulated in the parent study protocol [33]. Data collected using the android phones were temporarily stored in a secure and encrypted database on a memory card on the handheld device and uploaded onto an online server at the end of each day. The site office was secured to prevent theft and unwanted access to hardware and data. Access to collected data at the site was strictly controlled by the onsite data manager. Data were checked for completeness daily by the onsite data manager. Data collection commenced in January 2018 and stopped in March 2019 after the target sample size was reached. Other details of data management, quality control, randomization and masking have been published [33].

## Statistical analysis

Data were analyzed using Stata version 15.0 (StataCorp; College Station, TX, USA). Continuous variables were summarized into means, median, and standard deviation. Categorical variables were summarized into proportions. We compared baseline characteristics of the mother, household, and newborns between the intervention and the control groups to check for group comparability and identify potential confounders.

Analysis of the main outcome was based on intention-to-treat. The primary analysis compared the prevalence of mothers who initiated breastfeeding within the first hour, and who exclusively breastfed in the first 28 days of life, in the intervention and control group.

We calculated prevalence ratios for timely initiation of breastfeeding, and exclusive breastfeeding using a generalized estimating equation model for the Poisson family, with a log link, taking into account the clustering, and assuming an exchangeable correlation. This was done by a) trial arm and b) by cohort design; factors different across the two arms were entered into a multivariable generalized estimating equation model. We used Huber/white/sandwich variance estimators. Prevalence differences were calculated as above but using an identity link instead of a log link. We also calculated socioeconomic inequality using the Stata DASP package and we report Concentration Indexes. Finally, we assessed whether the intervention effect on timely breastfeeding was modified by the place of birth. This analysis was planned *a posteriori* to explore whether there was stronger evidence of the effect of the intervention on timely breastfeeding among those who gave birth at a health facility. To quantify any biological interaction between place of birth and the intervention, we estimated the absolute excess risk due to interaction by using Stata's icp command to fit a log-binomial model and we adjusted for clustering as shown in the Stata code below.

```
icp, show: binreg timely_breastfeeding intervention place_of_birth, nolog cformat(%7.3f) vce(cluster cluster) rr
```

## Results

### Trial profile

Following screening and recruitment into the trial [33], all 1877 pregnant women (Fig 1) were included in the current analysis. Nine hundred ninety-five (995) were in the intervention arm while 882 were in the control arm. Breastfeeding initiation status was assessed for 926 participants (93%) mothers in the intervention arm and 829 (94%) in the control arm. Exclusive

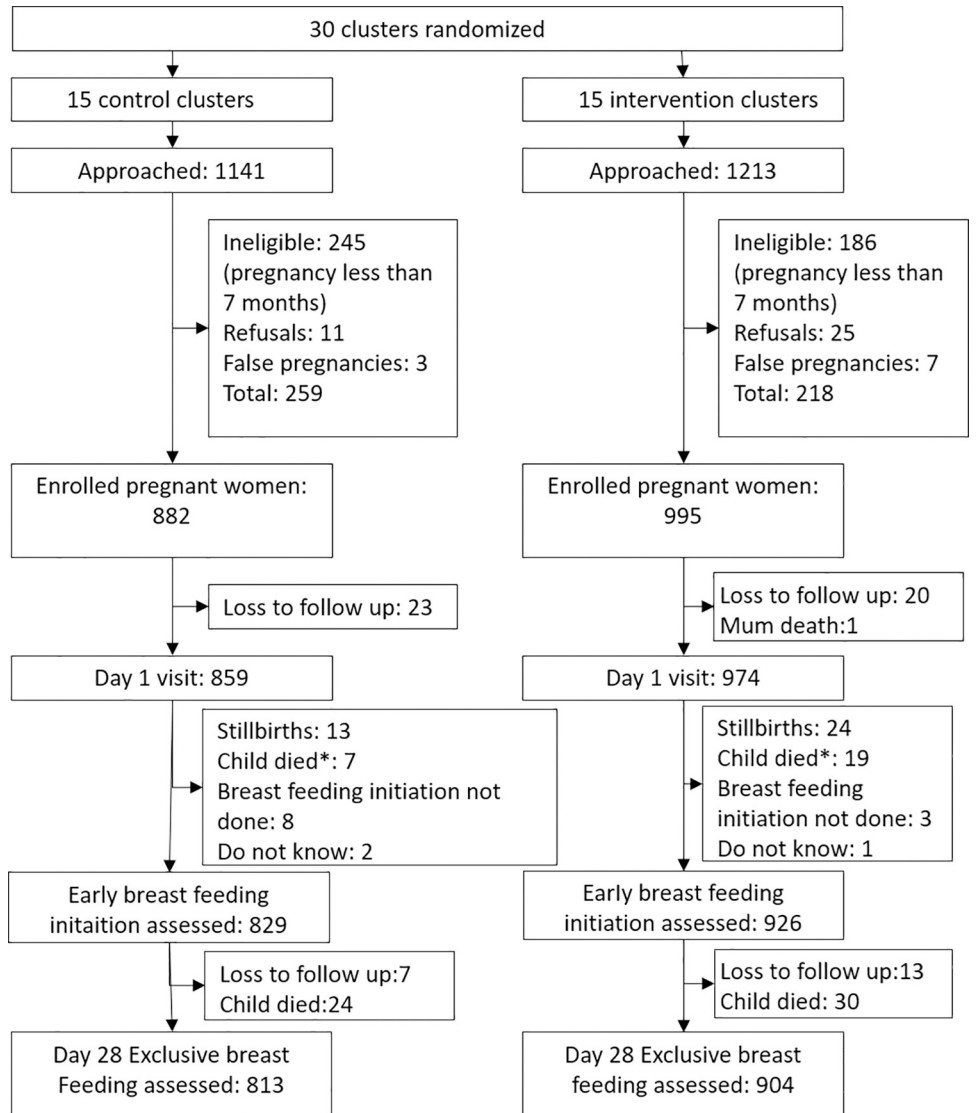

**Fig 1. Flow chart of participant recruitment in Northern Uganda [33].**

breastfeeding status was assessed for 904 participants (91%) mothers in the intervention arm and 813 (92%) in the control arm (Fig 1).

## Participant characteristics

Participants were largely similar between the two arms but we observed a few baseline differences. Sixteen percent (155/995) of the participants in the intervention arm had electricity in their homes compared to 6.2% (55/882) in the control arm (Table 1). Mothers in the intervention arm were more likely to give birth at a health facility, and possess mobile phones in their homesteads. Overall, only 55% (n = 1024) of participants owned a mobile phone in their household.

**Timely initiation of breastfeeding and exclusive breastfeeding.** Of participants in the intervention, 64% (594/926) initiated breastfeeding within the first hour after birth compared to 60% (493/829) in the control arm. Participants in the intervention arm were as likely to

**Table 1. Participant characteristics of mothers enrolled into a trial to assess for effectiveness of an integrated intervention package on timely initiation of and exclusive breastfeeding in Northern Uganda.**

| | Control N = 882 | Intervention N = 995 |
|---|---|---|
| **Continuous variables** | | |
| **Age of mother** | | |
| Mean (SD) | 24.4 (6.6) | 24.7 (6.9) |
| Median (IQR) | 23 (19–29) | 23 (19–29) |
| **Number of children mother has had before** | | |
| Mean (SD) | 3.1 (2.3) | 3.2 (2.3) |
| Median (IQR) | 3 (1–5) | 3 (1–5) |
| **Number of live children mother has at enrolment** | | |
| Mean (SD) | 2.7 (4.5) | 3.0 (2.2) |
| Median (IQR) | 2 (1–4) | 3 (1–5) |
| **Number of people living in household** | | |
| Mean (SD) | 4.7(2.4) | 4.9(2.6) |
| Median (IQR) | 4(3–6) | 4(3–6) |
| **Categorical variables** | n (%) | n (%) |
| **Marital status** | | |
| Married | 677 (76.8) | 790 (79.4) |
| Co-habiting | 110 (12.5) | 136 (13.7) |
| Single | 95 (10.8) | 69 (6.9) |
| **Mothers level of education** | | |
| None | 131 (14.9%) | 117 (11.8%) |
| Primary | 669 (75.9%) | 790 (79.4%) |
| > = Secondary | 82 (9.3%) | 88 (8.8%) |
| **Electricity** | | |
| Yes | 55 (6.2) | 155 (15.6) |
| **Presence of mobile phone in the household** | | |
| Yes | 442 (50.1) | 582 (58.5) |
| **Mother attended Antenatal Care Visits** | | |
| Yes | 696 (78.9) | 785 (78.9) |
| **Main provider of income in the household** | | |
| Father of the child you are carrying in womb | 770 (87.3%) | 872 (87.6%) |
| Mother | 47 (5.3%) | 49 (4.9%) |
| Older male relative | 14 (1.6%) | 29 (2.9%) |
| Older female relative | 14 (1.6%) | 16 (1.6%) |
| Other household member living at home | 5 (0.6%) | 5 (0.5%) |
| Other person | 32 (3.6%) | 24 (2.4%) |
| **Main occupation of husband** | | |
| Salaried worker | 39 (4.4%) | 61 (6.1%) |
| Businessman | 39 (4.4%) | 40 (4.0%) |
| Casual Laborer | 64 (7.3%) | 76 (7.6%) |
| Farmer | 602 (68.3%) | 677 (68.0%) |
| Other | 43 (4.9%) | 72 (7.2%) |
| **Wealth quintiles** | | |
| I (Poorest) | 202 (22.9) | 182 (18.3) |
| II | 203 (23.0) | 250 (25.1) |

*(Continued)*

**Table 1.** (Continued)

|  | Control N = 882 | Intervention N = 995 |
|---|---|---|
| III | 178 (20.2) | 167 (16.8) |
| IV | 132 (15.0) | 188 (18.9) |
| V (Richest) | 167 (18.9) | 208 (20.9) |
| **Place of Birth of Current Child (n = 1876)** |  |  |
| Health facility | 500 (56.7) | 754 (75.9) |
| Home | 382 (43.3) | 240 (24.1) |
| **Child characteristics** |  |  |
| **Sex (n = 1833)** |  |  |
| Male | 429 (49.9) | 484 (49.7) |
| **Low birthweight (n = 1523)** |  |  |
| Yes | 46 (6.4) | 45 (5.6) |
| **Multiple pregnancy** |  |  |
| Singleton | 869 (98.5%) | 987 (99.2%) |
| Multiple | 13 (1.5%) | 8 (0.8%) |
| **Born less than 37 completed weeks of gestation** |  |  |
| Yes | 118 (13.4%) | 122 (12.3%) |

initiate breastfeeding within the first hour of life, as participants in the control arm [Prevalence ratio (PR) 1.08 (0.97 to 1.21)] and [Prevalence difference (PD) 0.05 (-0.02 to 0.12)] (Table 2). In the intervention arm, 89% (804/904) of participants exclusively breastfed their infants in the first month of life compared to 81% (656/813) in the control arm. Participants in the intervention arm were 10% more likely to have exclusively breastfed in the preceding 24 hours compared to mothers in the control arm [PR 1.10 (1.04 to 1.17)] and [PD 0.08 (0.04 to 0.13)], and 16% more likely to have exclusively breastfed since birth compared to mothers in the control arm [PR 1.16 (1.03 to 1.30)] and [PD 0.12 (0.03 to 0.20] (Table 2).

When we adjusted for the presence of electricity and the presence of a mobile phone in the household, mothers in the intervention arm were as likely to have initiated breastfeeding within the first hour of life, and exclusively breastfed as before adjustment (Table 3).

The interaction term between the place of birth and the intervention was not significant. The relative excess risk due to interaction between the place of birth and the intervention was estimated to be 0.161 (95%CI: -0.034 to 0.356, p value = 0.1064).

Fewer mothers (32%, n = 555) reported being personally responsible for the timing of the first feed. Health workers initiated the timing of the first feed for forty percent (n = 683) of

**Table 2. Effect of the intervention on timely initiation of and exclusive breastfeeding in Northern Uganda before adjusting for the presence of electricity and phone in the home.**

|  | Control | Intervention | Prevalence Ratio (95% CI) | P value | Prevalence Difference (95% CI) |
|---|---|---|---|---|---|
| Timely breast feeding initiation | 493/ 829 (59.5%) | 594/926 (64.2%) | 1.08 (0.97 to 1.21) | 0.175 | 0.05 (-0.02 to 0.12) |
| Exclusive breast feeding (24hr recall) | 656/ 813 (80.7%) | 804/904 (88.9%) | 1.10 (1.04 to 1.17) | 0.001 | 0.08 (0.04 to 0.13) |
| Exclusive breast feeding (Monthly recall) | 584/813 (71.8%) | 761/904 (84.2%) | 1.16 (1.03 to 1.30) | 0.013 | 0.12 (0.03 to 0.20) |

**Table 3.** Effect of the intervention on timely breastfeeding initiation and exclusive breastfeeding in Northern Uganda adjusting for the presence of electricity and phone in the home.

|  | Adjusted Prevalence Ratio (95% CI) | P value |
|---|---|---|
| **Timely breastfeeding initiation** | | |
| Intervention | 1.06 (0.95 to 1.19) | 0.291 |
| Electricity in home | 1.14 (1.0 to 1.3) | 0.047 |
| Phone in home | 1.07 (1.0 to 1.1) | 0.034 |
| **Exclusive breast feeding (24hr recall)** | | |
| Intervention | 1.11 (1.04 to 1.17) | 0.001 |
| Electricity in home | 1.01 (0.93 to 1.09) | 0.888 |
| Phone in home | 0.98 (0.95 to 1.02) | 0.305 |
| **Exclusive breast feeding (1 month recall)** | | |
| Intervention | 1.16 (1.03 to 1.30) | 0.014 |
| Electricity in home | 1.06 (1.00 to 1.12) | 0.080 |
| Phone in home | 0.97 (0.94 to 1.00) | 0.080 |

mothers. When we restricted the analysis to only mothers who decided on when to breastfeed, there was some evidence of intervention effectiveness [PR 1.20, 95% CI (0.99 to 1.5)] (Table 4, Figs 2 and 3). Only 55% (n = 1024) of homesteads had a mobile phone (Table 1). Timely breastfeeding initiation was pro-rich [Concentration Index 0.045, 95% CI (0.011 to 0.079)] but the intervention did not decrease this inequality (Table 4, Figs 2 and 3). There was almost no inequality concerning exclusive breastfeeding (Table 4, Figs 2 and 3). The intra-cluster correlation coefficient for timely initiation of breastfeeding was 0.03, 95% CI (0.01 to 0.07), whereas the intra-cluster correlation coefficient for exclusive breastfeeding was 0.07, 95% CI (0.03 to 0.15).

## Discussion

The intervention increased the proportion of women who exclusively breastfed at one month postpartum but did not change the proportion of women who initiated breastfeeding within the first hour after birth.

Sixty-four percent of newborns in the intervention clusters were initiated to breastfeeding in the first one hour of life, compared to 60% in the control clusters. Studies in other settings

**Table 4.** Proportions by inequalities and concentration indexes of timely initiation of and exclusive breastfeeding in Northern Uganda.

|  | N | Q1 (%) | Q2 (%) | Q3 (%) | Q4 (%) | Q5 (%) | Mean | Concentration Index | 95% CI |
|---|---|---|---|---|---|---|---|---|---|
| **Timely breast-feeding initiation** | | | | | | | | | |
| Control | 829 | 52.1 | 58.2 | 64.7 | 57 | 66.5 | 59.5 | 0.045 | 0.011 to 0.079 |
| Intervention | 926 | 60.6 | 63.7 | 64.9 | 67.4 | 64.3 | 64.2 | 0.012 | −0.025 to 0.048 |
| Total | 1755 | 56.1 | 61.2 | 64.8 | 63.2 | 65.2 | 61.9 | 0.028 | 0.002 to 0.053 |
| **Exclusive breast feeding (24hr recall)** | | | | | | | | | |
| Control | 813 | 79.9 | 81.4 | 82.1 | 78.6 | 80.9 | 80.7 | 0 | −0.020 to 0.020 |
| Intervention | 904 | 89.8 | 89.5 | 86.2 | 92.3 | 86.8 | 88.9 | −0.004 | 0.020 to 0.011 |
| Total | 1717 | 84.5 | 85.8 | 84.1 | 86.7 | 84.2 | 85 | −0.001 | −0.013 to 0.011 |
| **Exclusive breast feeding (1 Month recall)** | | | | | | | | | |
| Control | 813 | 73 | 70 | 69.8 | 70.1 | 76.3 | 71.8 | 0.005 | 0.022 to 0.032 |
| Intervention | 904 | 84.9 | 83.8 | 83.6 | 84.5 | 84.1 | 84.2 | −0.002 | −0.017 to 0.012 |
| Total | 1717 | 78.6 | 77.5 | 76.4 | 78.6 | 80.7 | 78.3 | 0.003 | −0.011 to 0.017 |

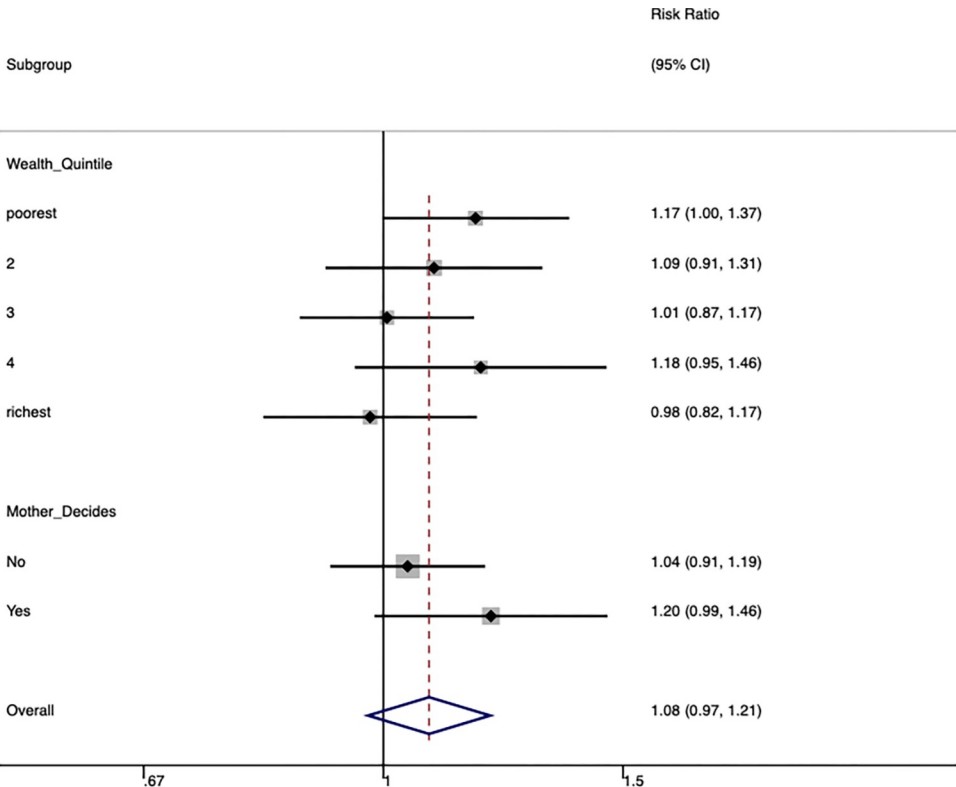

**Fig 2. Sub-group analysis showing the effect of intervention on early breastfeeding initiation by wealth index and mother's decision regarding early breastfeeding initiation in Northern Uganda.**

have observed similar findings [35–38]. The absence of a significant difference could partly be due to the fact that our intervention was only focused on pregnant women and their families. In a review of interventions that improve breastfeeding outcomes, Sinha *et al*. noted that breastfeeding interventions that combined health systems, home, family and community settings had the greatest impact [39,40]. Other authors have also come to the same conclusion [41]. The absence of a significant effect of the intervention on the proportion of mothers who initiated breastfeeding within the first hour of life could also have been due to the absence of decision-making power by mothers in the immediate postpartum period. Health care providers are often the main decision-makers in the immediate postpartum period, and little can be done without their cooperation. Participants in our study were asked who was responsible for the time they took to initiate breastfeeding, and approximately 40% of them stated that it was the health workers who decided when they initiated breastfeeding. Since most of the births 67% (1254/1876) happened in hospital, health system factors could have affected time of initiation of breast feeding.

Sub-group analysis of those who reported that they decided when to initiate breastfeeding showed some evidence that the intervention was effective. This highlights the need to emphasize maternal autonomy in regards to reproductive and child health. Women need to be empowered and involved in making decisions regarding their health to ensure adherence to recommendations such as early initiation of breast feeding. Maternal autonomy results in

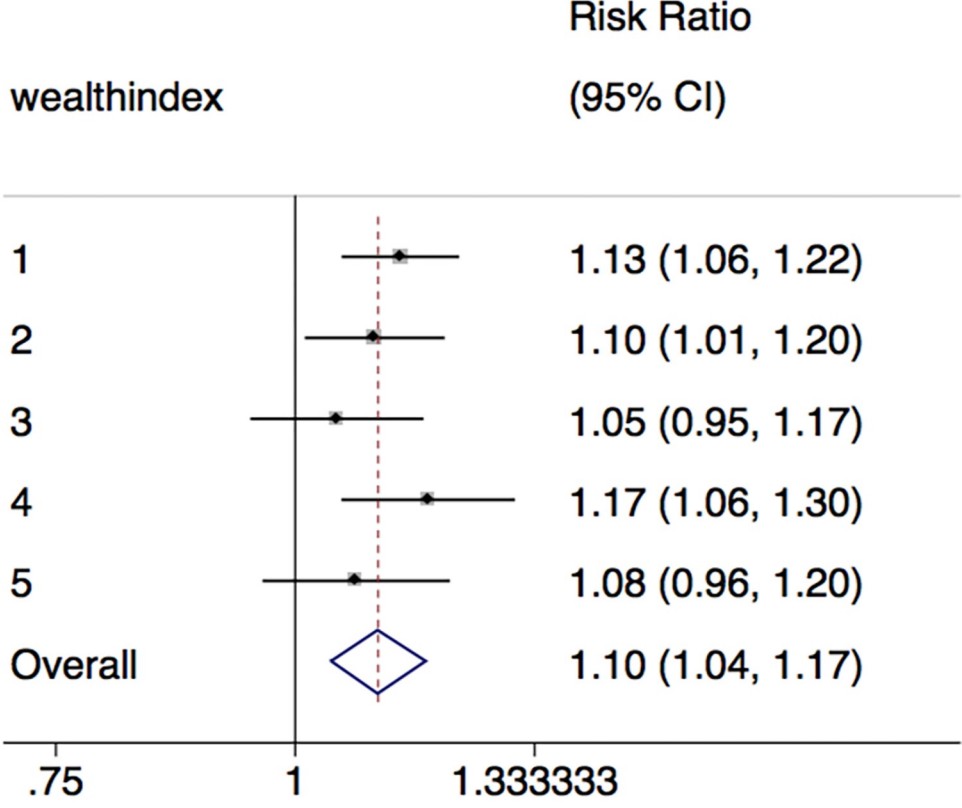

**Fig 3. Subgroup analysis showing the effect of the intervention on exclusive breastfeeding (measured by 24-hour recall) by wealth index.**

intrinsic motivation and behavioral regulation, which brings about positive, sustained, and long-term behavioral change [42].

Our findings also suggest that health service providers need to be involved when designing interventions that promote timely initiation of breastfeeding [39,40]. Mothers often experience anxiety in the immediate postpartum period and may not be able to recall the breastfeeding instructions given to them during the antenatal period. To address this, our intervention had also involved significant others like husbands and co-wives. However, in Uganda, like most other low-income countries, husbands and other relatives do not have access to the labor ward and are often absent during the birthing process. This could also partly explain why the intervention did not improve early breastfeeding initiation.

Other investigators have however observed a significant improvement in the prevalence of timely initiation with peer counselors [43–45]. A study conducted in Uganda, Burkina Faso, and South Africa showed that peer counseling improved timely initiation of breastfeeding in Uganda, but not in Burkina Faso or South Africa. These trials were solely conducted to promote breastfeeding practices and did not have any other interventions, and this might have ensured better trial fidelity [44]. However, in low and middle-income settings, intervention packages are proposed as an efficient and effective way to deliver care [46] and we had hypothesized that utilizing peer counselors to encourage both health facility births, and newborn feeding practices would be more pragmatic and cost-effective.

On the other hand, our intervention increased the prevalence of exclusive breastfeeding. This finding was not surprising since studies done in Eastern Uganda [28], and elsewhere in the world [29,45,47,48] have demonstrated the effectiveness of community-based interventions in the promotion of exclusive breastfeeding. The effectiveness of our intervention in promoting exclusive breastfeeding but not timely initiation of breastfeeding could have been caused by the greater control a mother has over what she can do with her baby in the later postnatal period, compared to the immediate postnatal period [49].

### Strengths and limitations

This was a community study, in an area where home births constitute a third of all births. An effort was made to approach mothers within the first 24 hours after birth, and 75% of mothers were interviewed in the first 48 hours, and 90% were visited within the first 4 days after birth; which is a major strength as the recall limitation was reduced. However, it is still possible that misclassification, as a result of poor maternal recall, may have affected the outcome measurement.

A major limitation to this study is that the main outcome of the study was a secondary objective of our trial, which could have negatively affected trial fidelity, as peer counselors were taught to prioritize the health facility births message. However, when we re-interviewed participants in the intervention group to assess study fidelity and quality, 766/805 (95.2%) of the participants in the intervention arm reported that a peer counsellor visited them. Of the participants that were visited by a peer counsellor, 707/766 [92.3%: (95%CI 89.0–94.6)] reported that they had been counselled on early breastfeeding initiation. When asked after how long time after birth a baby should ideally be put to the breast, 747/766 [97.3%: (95% CI 95.3–98.4)] gave an answer within an hour suggesting sufficient knowledge. The effectiveness of the intervention in promoting exclusive breastfeeding hints at the possibility of using peer counselors to deliver packages, rather than solitary interventions. Multi-focus interventions are better for the planning and financing of health systems [50]. The decision to conduct a sub-group analysis on mothers who reported that they made the decision when to initiate breastfeeding was made a *posteriori* and therefore the results should be treated with caution.

Another limitation to the implementation of the study was the fact that only 55% of mothers had mobile phones, and this could have limited the number of people who received the messages. We did not access health workers' knowledge and practices regarding timely breastfeeding initiation and health facility barriers regarding timely breastfeeding initiation. Finally, a qualitative exploration of the decision-making and barriers surrounding timely breastfeeding in health facilities could have added clarity to our results, but this was not done.

## Conclusion

The intervention improved the proportion of mothers who practiced exclusive breastfeeding in the first month of life but did not increase the proportion of mothers who initiated breastfeeding in the first hour of life. Timely breastfeeding initiation was more prevalent among richer households but the intervention did not decrease this inequality. Future breastfeeding promotion interventions should consider including a health facility component and improving maternal autonomy to promote timely initiation of breastfeeding.

### Patient and public involvement

The public was not involved in the design and conceptualization of the study, but they were involved in the recruitment of participants. We held community meetings in each village during which a recruiter was elected from among the village members. The recruiter was

responsible for recruitment in their village. The results of this study will be disseminated to the wider community through community dialogue meetings at the parish level in each participating village.

## Supporting information

**S1 File. Consort checklist.**
(DOC)

**S2 File. Anonymized dataset.**
(XLSX)

**S3 File. Anonymized codebook.**
(XLS)

**S4 File. Study protocol.**
(PDF)

**S5 File. Map of study area.**
(PDF)

## Acknowledgments

We acknowledge the Survival Pluss project (no. UGA-13-0030) at Makerere University. In a special way, we acknowledge the District Health Office of Lira district, and the various district, sub-county, parish and village leaders for their assistance in this study. Finally, we acknowledge the participants and research assistants in our study for their cooperation.

## Author Contributions

**Conceptualization:** David Mukunya, James K. Tumwine, Grace Ndeezi, Thorkild Tylleskar, Victoria Nankabirwa.

**Data curation:** David Mukunya.

**Formal analysis:** David Mukunya, James K. Tumwine, Faith Oguttu, Thorkild Tylleskar.

**Funding acquisition:** James K. Tumwine, Grace Ndeezi, Thorkild Tylleskar.

**Investigation:** David Mukunya, James K. Tumwine, Grace Ndeezi, Milton W. Musaba, Justin Bruno Tongun, Josephine Tumuhamye, Agnes Napyo, Victoria Nankabirwa.

**Methodology:** David Mukunya, James K. Tumwine, Milton W. Musaba, Justin Bruno Tongun, Josephine Tumuhamye, Agnes Napyo, Faith Oguttu, Daphine Amanya, Beatrice Odongkara, Vincentina Achora.

**Project administration:** David Mukunya, James K. Tumwine.

**Supervision:** David Mukunya, James K. Tumwine, Grace Ndeezi, Beatrice Odongkara, Vincentina Achora, Thorkild Tylleskar, Victoria Nankabirwa.

**Writing – original draft:** David Mukunya, James K. Tumwine, Faith Oguttu, Thorkild Tylleskar.

**Writing – review & editing:** David Mukunya, Grace Ndeezi, Milton W. Musaba, Justin Bruno Tongun, Josephine Tumuhamye, Agnes Napyo, Faith Oguttu, Daphine Amanya, Beatrice Odongkara, Vincentina Achora, Victoria Nankabirwa.

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
