## [Decision Letter · Decision Letter 0]

2 Sep 2024

PONE-D-24-18566Assessing a bundle of peer counseling, mobile phone messages, and mama kits in promoting timely initiation of and exclusive breastfeeding in Uganda: a cluster randomized controlled studyPLOS ONE

Dear Dr. Mukunya,

Thank you for submitting your manuscript to PLOS ONE. After careful consideration, we feel that it has merit but does not fully meet PLOS ONE’s publication criteria as it currently stands. Therefore, we invite you to submit a revised version of the manuscript that addresses the points raised during the review process.**Ensure all abbreviations are explained, and figures/tables are clearly labeled. Consistent statistical language will strengthen the manuscript.****Consider adding broader context on public health interventions, especially regarding reproductive rights and maternal autonomy.****Ensure your dataset is complete and aligned with the manuscript. Include a codebook for clarity.as asuplimentary file****Include more specific information about the intervention content, including the mobile messages and peer counseling components. This will enhance reproducibility and allow others to better understand and replicate the study.****Justify the choice of statistical models used in your analysis. Additionally, consider mentioning any sensitivity analyses conducted to assess the robustness of your findings.****The methodology section is detailed, but there is a need for more information on how participant characteristics (e.g., presence of electricity and mobile phones) were controlled for in the analysis. Please provide a more thorough explanation of the statistical methods used to account for these socio-economic factors.****The results are presented clearly, but the interpretation of the findings could be strengthened. Specifically, discuss why the intervention did not significantly impact timely breastfeeding initiation despite improvements in exclusive breastfeeding rates. Are there other factors that might have influenced these outcomes?****The figures and tables are informative but might benefit from additional explanatory notes. Ensure that all tables are referenced in the text and that figures are accompanied by captions that clearly describe what is being presented****The paper is generally well-written, but there are a few areas where language and grammar could be improved for clarity. For instance, review the use of passive voice and ensure consistent terminology throughout the paper**Please submit your revised manuscript by Oct 17 2024 11:59PM. If you will need more time than this to complete your revisions, please reply to this message or contact the journal office at plosone@plos.org. Please include the following items when submitting your revised manuscript:A rebuttal letter that responds to each point raised by the academic editor and reviewer(s). You should upload this letter as a separate file labeled 'Response to Reviewers'.A marked-up copy of your manuscript that highlights changes made to the original version. You should upload this as a separate file labeled 'Revised Manuscript with Track Changes'.An unmarked version of your revised paper without tracked changes. You should upload this as a separate file labeled 'Manuscript'.If applicable, we recommend that you deposit your laboratory protocols in protocols.io to enhance the reproducibility of your results. Protocols.io assigns your protocol its own identifier (DOI) so that it can be cited independently in the future. For instructions see: https://journals.plos.org/plosone/s/submission-guidelines#loc-laboratory-protocols. Additionally, PLOS ONE offers an option for publishing peer-reviewed Lab Protocol articles, which describe protocols hosted on protocols.io. Read more information on sharing protocols at https://plos.org/protocols?utm_medium=editorial-email&utm_source=authorletters&utm_campaign=protocols.

We look forward to receiving your revised manuscript.

Kind regards,

Trhas Tadesse Berhe, PhD

Academic Editor

PLOS ONE

Journal Requirements:

We acknowledge the Survival Pluss project (no. UGA-13-0030) at Makerere University. Survival Pluss project was funded under NORHED by Norad; Norway.

We acknowledge the Survival Pluss project (no. UGA-13-0030) at Makerere University. Survival Pluss project was funded under NORHED by Norad; Norway.

In a special way, we acknowledge the District Health Office of Lira district, and the various district, sub-county, parish and village leaders for their assistance in this study. Finally, we acknowledge the participants and research assistants in our study for their cooperation.

We acknowledge the Survival Pluss project (no. UGA-13-0030) at Makerere University. Survival Pluss project was funded under NORHED by Norad; Norway.

Reviewers' comments:

Reviewer's Responses to Questions

**Comments to the Author**

1. Is the manuscript technically sound, and do the data support the conclusions?

Reviewer #1: Yes

Reviewer #2: Yes

Reviewer #3: Yes

Reviewer #4: Yes

2. Has the statistical analysis been performed appropriately and rigorously? 

Reviewer #1: Yes

Reviewer #2: Yes

Reviewer #3: Yes

Reviewer #4: Yes

3. Have the authors made all data underlying the findings in their manuscript fully available?

Reviewer #1: Yes

Reviewer #2: Yes

Reviewer #3: Yes

Reviewer #4: No

4. Is the manuscript presented in an intelligible fashion and written in standard English?

Reviewer #1: Yes

Reviewer #2: Yes

Reviewer #3: Yes

Reviewer #4: Yes

5. Review Comments to the Author

**Reviewer #1:** Thank you for the possibility to read a well-executed study and a well-written manuscript.

I have some minor comments.

Figure 1 Trial profile: Please add which group is the intervention group. Please also explain LTFU, EBF, and EBI.

Line 293: Is it a supplementary table?

Table 1: You don’t need to report the No, when you report the Yes (electricity, mobile phone, antenatal care visits, sex, LBW).

Table 2, 3 and 4: Please explain the abbreviation in the tables (even though it is explained in the text) (BFI, BF, EBI, EBF)

Line 326-328: Please optimize the sentence for better reading. (e.g: …; fewer mothers (32%, n=555) reported they themselves were responsible for the timing of the first feed).

Please discus in the limitation section the 55% coverage of mobile phones when bundle included mobile phone messages.

Line 338: The title of Table 4 is beneath the table (it is above the other three tables).

Figure 2: Please name “0” and “1” (Motherdecides). Make sure the figure has a title.

**Reviewer #2**: A cluster randomized trial was conducted to evaluate the impact of an integrated intervention consisting of peer counseling, mobile phone messages, and mama kits on increasing health facility births. Additionally, the study aimed to examine the effect of the intervention on promoting timely initiation of exclusive breastfeeding. The proportion of participants in the intervention initiating breastfeeding within the first hour of life did not significantly differ from that in the control group. Moreover, mothers in the intervention group showed a 10% higher likelihood of exclusively breastfeeding in the preceding 24 hours, and a 16% higher likelihood of exclusively breastfeeding since birth compared to mothers in the control group.

Minor revisions:

1- In the abstract, rephrase the sentence containing the words "equally likely" to reflect the more statistically accurate phraseology provided in the following statement. The proportion of participants in the intervention initiating breastfeeding within the first hour of life did not significantly differ from that in the control group.

2- Supplementary Table 1: Provide p-values comparing the control to the intervention arm.

3- Table 2: State the statistical testing method used to estimate the 95% confidence intervals.

4- The standard statistical term for average is mean.

**Reviewer #3:** A few points to consider:

Abstract:

Line 30 - if this is the target- you may want to state what the current norm is.

Introduction

Line 69- these numbers are different to what you have quoted in your abstract

Line 81- it is not just the child that initiates or controls breastfeeding, i think the author is trying to say, children who are breastfed early, not 'initiate'.

Results

Line 287-288- Is there any explanation as to why mothers in the intervention arm were far more likely to have electricity than the control arm or is this by chance?

**Reviewer #4:** The authors conducted a cluster randomized controlled trial in a district in Northern Uganda during 2018-2019 to evaluate the impact of a package of interventions (peer counseling, mobile phone messages, and mama kits) designed to promote maternal and child health. The primary objective of the intervention—encouraging births at health facilities—was previously published (BMJ Open, 2024, cited as reference 33 in the submitted paper). This new paper focuses on assessing the intervention's effect on two secondary objectives related to breastfeeding.

INTRODUCTION

The first part of the introduction effectively highlights the significance of this research (lines 68 to 113). However, the second part (lines 114 to 137) would benefit from revisions for clarity, with some elements better suited for the Methods section. The introduction should conclude with a clear statement of the paper's objective.

METHODS

--------------

The paper presents a generally robust methodology.

However, the “sample size” section (lines 193-204) is somewhat unconvincing, as it’s difficult to believe that the sample size was estimated without a specific target below 1,877 participants. I recommend stating upfront that the sample size was calculated based on the primary objective (rather than mentioning this at the end of the section). Additionally, it would be more transparent to explain that the authors used Stata IC to verify what could be achieved with the given sample size.

The authors reference a “baseline survey” (lines 130 & 197). Could you provide a brief description of this survey?

Three outcomes were investigated: (1) initiation of breastfeeding within the first hour after birth, (2) exclusive breastfeeding in the 24 hours before the visit 28 days after birth, and (3) exclusive breastfeeding during the first 28 days of life. While the first outcome is detailed (lines 207-214), the other two are only briefly mentioned (lines 214-217). These outcomes should be expanded upon with clear definitions in the Methods section, along with information on data collection: When did the 28-day visit actually take place? How were the two outcomes measured?

In the Methods section, please clarify the “socio-economic status quintiles” variable. Was it based solely on wealth? How was wealth assessed? Additional information on the definitions of other baseline characteristics and potential confounders (lines 218-223) would also be helpful.

Could you also explain how you collected information on “who decided when to initiate breastfeeding”? This is a critical variable in your analysis and should be thoroughly detailed in the Methods section.

RESULTS

--------------

The paper offers a rigorous analysis.

"Supplementary Table 1" is a key table and should be included in the main manuscript rather than as supplementary material. Several important baseline characteristics and potential confounders are missing from Supplementary Table 1 and should be added: maternal education, occupation, gestational age, household size, and whether the birth was singleton or multiple. Additionally, including confidence intervals (CIs) or p-values would help interpret the differences between the two groups.

DISCUSSION AND CONCLUSION

The authors concluded that the intervention improved exclusive breastfeeding but had no significant impact on the initiation of breastfeeding within the first hour after birth. These conclusions align with the statistical analysis results.

While the discussion is appropriate, it could be strengthened by offering a broader perspective on public health interventions in maternal health. Specifically, discussing this topic in the context of reproductive rights and maternal autonomy would add valuable depth.

The authors hypothesize (lines 353-354) that the lack of effect on breastfeeding initiation within the first hour may be due to this timing being more influenced by health workers at the facility than by the mother. This is an important point that warrants further exploration: Should health workers receive additional training on the importance of early breastfeeding initiation? Are there organizational challenges within health facilities that need addressing? This perspective is crucial for future research and public health action and should be developed more fully.

DATA AVAILABILITY

The study’s database has been submitted (Excel file) in accordance with PLOS Data Policy, but it requires revisions and additions:

- The number of observations: n=1898. This includes empty observations and those with only one data point (mum_decide=0). The number of observations should match the number of enrolled pregnant women (n=1877).

- Additional variables should be included in the database, as it's currently not possible to verify Figure 1 (e.g., “stillbirths,” “child died,” “LTFU”). Additionally, variables such as maternal education and occupation are missing from the Excel file.

- A codebook should accompany the database, detailing the variable names and codes.

- Please review the coding of variables. For instance, the “marital status” variable needs revision: it is listed as a two-category variable (married/single) in the database but is analyzed as a three-category variable (Supplementary Table 1—married/cohabiting/single).

MINOR COMMENTS

- Line 95: Instead of “less than two-thirds of women in sub-Saharan,” please provide the exact percentage.

- Line 96: Instead of “two-thirds of mothers in Uganda,” please provide the exact percentage.

- Line 141: The “map of the study villages in appendix 1” is missing.

- Figure 1: Please clarify all abbreviations used in the figure (LTFU, EBI, EBF).

- Line 430-431: The phrase “Timely breastfeeding initiation was pro-rich” needs revision for clarity.

6. PLOS authors have the option to publish the peer review history of their article (what does this mean?). If published, this will include your full peer review and any attached files.

Reviewer #1: No

Reviewer #2: No

Reviewer #3: No

Reviewer #4: No

---

## [Author Response · Author response to Decision Letter 0]

15 Oct 2024

We have included all responses in our response letter and the cover letter as advised.

---

## [Decision Letter · Decision Letter 1]

15 Nov 2024

PONE-D-24-18566R1Assessing a bundle of peer counseling, mobile phone messages, and mama kits in promoting timely initiation of and exclusive breastfeeding in Uganda: a cluster randomized controlled studyPLOS ONE

Dear Dr. David Mukunya

Thank you for submitting your manuscript to PLOS ONE. After careful consideration, we feel that it has merit but does not fully meet PLOS ONE’s publication criteria as it currently stands. Therefore, we invite you to submit a revised version of the manuscript that addresses the points raised during the review process.

Kindly  review and address the comments provided by the fifth  reviewer in detail. Ensure that their concerns are thoroughly considered and reflected in your revisions. If any suggestions cannot be implemented, provide a clear and justified explanation for your decision.

We look forward to receiving your revised manuscript.

Kind regards,

Trhas Tadesse Berhe, PhD

Academic Editor

PLOS ONE

Journal Requirements:

Reviewers' comments:

Reviewer's Responses to Questions

**Comments to the Author**

1. If the authors have adequately addressed your comments raised in a previous round of review and you feel that this manuscript is now acceptable for publication, you may indicate that here to bypass the “Comments to the Author” section, enter your conflict of interest statement in the “Confidential to Editor” section, and submit your "Accept" recommendation.

Reviewer #2: All comments have been addressed

Reviewer #5: (No Response)

2. Is the manuscript technically sound, and do the data support the conclusions?

Reviewer #2: (No Response)

Reviewer #5: Yes

3. Has the statistical analysis been performed appropriately and rigorously? 

Reviewer #2: (No Response)

Reviewer #5: Yes

4. Have the authors made all data underlying the findings in their manuscript fully available?

Reviewer #2: (No Response)

Reviewer #5: Yes

5. Is the manuscript presented in an intelligible fashion and written in standard English?

Reviewer #2: (No Response)

Reviewer #5: Yes

6. Review Comments to the Author

Reviewer #2: (No Response)

Reviewer #5: As mentioned, 40% of the time it was healthcare worker taking decision for the first hour breastfeeding. I am curious to know, why would a healthcare worker not advice in favor of early initiation. Is it due to inadequate staffing to assist mother for early initiation? or some other reason?

I wish to understand the reason behind failure to initiate early breastfeeding. Did you try to explore them?

7. PLOS authors have the option to publish the peer review history of their article (what does this mean?). If published, this will include your full peer review and any attached files.

Reviewer #2: No

Reviewer #5: **Yes: **Akanksha Jain

---

## [Author Response · Author response to Decision Letter 1]

20 Nov 2024

Unfortunately, we did not explore the reasons behind this phenomenon. For sure, we have inadequate staffing in these health facilities. Each shift has only one midwife on duty, and they often attend to about 10 women in labour. As such, supporting a woman to initiate breastfeeding within the first hour becomes a secondary task that is attended to once all other tasks are done. Also, we cannot rule out a lack of knowledge and poor attitudes towards timely breastfeeding. These are heavy accusations that we did not systematically explore. We have added this is a limitation.

---

## [Decision Letter · Decision Letter 2]

23 Dec 2024

Assessing a bundle of peer counseling, mobile phone messages, and mama kits in promoting timely initiation of and exclusive breastfeeding in Uganda: a cluster randomized controlled study

PONE-D-24-18566R2

Dear Dr. David Mukunya,

We’re pleased to inform you that your manuscript has been judged scientifically suitable for publication and will be formally accepted for publication once it meets all outstanding technical requirements.

Kind regards,

Trhas Tadesse Berhe, PhD

Academic Editor

PLOS ONE

Additional Editor Comments (optional):

Reviewers' comments:

Reviewer's Responses to Questions

**Comments to the Author**

1. If the authors have adequately addressed your comments raised in a previous round of review and you feel that this manuscript is now acceptable for publication, you may indicate that here to bypass the “Comments to the Author” section, enter your conflict of interest statement in the “Confidential to Editor” section, and submit your "Accept" recommendation.

Reviewer #2: All comments have been addressed

Reviewer #6: All comments have been addressed

2. Is the manuscript technically sound, and do the data support the conclusions?

Reviewer #2: (No Response)

Reviewer #6: Yes

3. Has the statistical analysis been performed appropriately and rigorously? 

Reviewer #2: (No Response)

Reviewer #6: Yes

4. Have the authors made all data underlying the findings in their manuscript fully available?

Reviewer #2: (No Response)

Reviewer #6: Yes

5. Is the manuscript presented in an intelligible fashion and written in standard English?

Reviewer #2: (No Response)

Reviewer #6: Yes

6. Review Comments to the Author

Reviewer #2: (No Response)

Reviewer #6: Accept, accept, accept! Excellent project and write-up of same. Since there was only one tiny addition, I have made no additional comments.

7. PLOS authors have the option to publish the peer review history of their article (what does this mean?). If published, this will include your full peer review and any attached files.

Reviewer #2: No

Reviewer #6: No

---

## [Editor Report · Acceptance letter]

13 Jan 2025

PONE-D-24-18566R2 

PLOS ONE

Dear Dr. Mukunya, 

I'm pleased to inform you that your manuscript has been deemed suitable for publication in PLOS ONE. Congratulations! Your manuscript is now being handed over to our production team.

Kind regards, 

on behalf of

Dr. Trhas Tadesse Berhe 

Academic Editor

PLOS ONE